# Physical Frailty and Oral Frailty Associated with Late-Life Depression in Community-Dwelling Older Adults

**DOI:** 10.3390/jpm12030459

**Published:** 2022-03-14

**Authors:** Ying-Chun Lin, Shan-Shan Huang, Cheng-Wei Yen, Yuji Kabasawa, Chien-Hung Lee, Hsiao-Ling Huang

**Affiliations:** 1Department of Dentistry, Kaohsiung Medical University Hospital, Kaohsiung 80708, Taiwan; bonnie0925.tw@gmail.com (Y.-C.L.); 1070480@kmuh.org.tw (S.-S.H.); 1040474@kmuh.org.tw (C.-W.Y.); 2Department of Oral Hygiene, College of Dental Medicine, Kaohsiung Medical University, Kaohsiung 80708, Taiwan; 3Oral Care for Systemic Health Support, Faculty of Dentistry, School of Oral Health Care Sciences, Graduate School, Tokyo Medical and Dental University, Tokyo 113-8510, Japan; kabasawa.ocsh@tmd.ac.jp; 4Department of Public Health, College of Health Sciences, Kaohsiung Medical University, Kaohsiung 80708, Taiwan; cnhung@kmu.edu.tw

**Keywords:** frailty, depression, oral health, dysphagia, xerostomia, sarcopenia, insomnia, older adults

## Abstract

Late-life depression is a major mental health problem and constitutes a heavy public health burden. Frailty, an aging-related syndrome, is reciprocally related to depressive symptoms. This study investigated the associations of physical frailty and oral frailty with depression in older adults. This large-scale cross-sectional study included 1100 community-dwelling older adults in Taiwan. The participants completed a dental examination and questionnaires answered during personal interviews. The 15-item Geriatric Depression Scale was used to assess depression, and information on physical conditions and oral conditions was collected. Multivariable logistical regression analysis was conducted to examine associations of interest. Significant factors associated with depression were pre-physical frailty (adjusted odds ratio (aOR) = 3.61), physical frailty (aOR = 53.74), sarcopenia (aOR = 4.25), insomnia (aOR = 2.56), pre-oral frailty (aOR = 2.56), oral frailty (aOR = 4.89), dysphagia (aOR = 2.85), and xerostomia (aOR = 1.10). Depression exerted a combined effect on physical frailty and oral frailty (aOR = 36.81). Physical frailty and oral frailty were significantly associated with late-life depression in community-dwelling older adults in a dose–response manner. Developing physical and oral function interventions to prevent depression among older adults is essential.

## 1. Introduction

The aging population is growing worldwide. At present, 16% of Taiwanese are older than 65 years. Ten years from now, older adults are projected to constitute one-fourth of Taiwan’s population [1]. Depression is not a standard element of the aging process, but it is a major mental health problem and a major public health concern among older adults [2]. As of 2020, depression is the second greatest disease burden [3]. In Taiwan, the rate of suspected depression in older adults is 13.3% [4]. Blazer [5] reported that behavioral, psychodynamic, and cognitive aberrations have all been suggested as causes of late-life depression. The late-life depression is associated with increased risk of morbidity, increased risk of suicide, impaired physical, cognitive, and social functioning, and greater self-neglect, all of which are in turn associated with increased mortality [5].

The late-life depression is part of frailty. Frailty is caused by life-course determinants and disease(s), accumulation of physical, psychological and/or social deficits in functioning, increasing the risk of negative health outcomes such as disabilities, admission to health care facilities, and death, is increasingly common in older adults [6]. Jung et al. [7] reported that physiconutritional (e.g., malnutrition, sarcopenia, severe mobility limitation et al.), psychological, sociodemographic, and medical factors were strongly associated with frailty. Sarcopenia, a key component of physical frailty, is characterized by the gradual decline of muscle mass and strength; moreover, it can aggravate swallowing disorders in older adults [8]. Sarcopenic muscles dedicated to the swallowing mechanism (e.g., tongue, geniohyoid, and pharyngeal muscles) can alter deglutition and potentially place an individual at risk for aspiration and aspiration pneumonia [9]. Tanaka et al. [10] observed a significant association between impaired oral function and the pathogenesis of frailty; difficulty eating leads to physical decline and functional disability and contributes to frailty.

Depression has been found to be strongly associated with oral function [11,12] and xerostomia [13] in older adults. Deterioration in oral health in the elderly is most often related to somatic disorders and polypharmacy [14]. Medication-induced xerostomia is common among older adults, especially when they take antidepressants [14]. A meta-analysis confirmed that eating a plant-based diet, wherein the consumption of vegetables, fruits, legumes and whole grains is high, is associated with a lower risk of depression [15]. However, tooth loss, a common problem among older adults with fewer than two posterior occlusal support areas (POSAs), causes impaired masticatory performance, which in turn negatively affects their ability to chew certain foods, particularly fruits and vegetables [16]. Aging can compromise food oral processing if it entails a decline in dental status and changes in oral physiology (e.g., mastication performance and swallowing problems) [17]. This condition is defined as oral frailty and can limit food intake [18]. The resulting poor nutritional status increases the risk of physical frailty [10] and depression [19].

Studies have demonstrated an association between frailty and depression [7,20]. Associations of the combined effects of physical frailty and oral frailty with depression remain unclear. This study aimed to investigate associations of physical frailty and oral frailty with depression in community-dwelling older adults.

## 2. Materials and Methods

### 2.1. Study Design and Participants

This large-scale cross-sectional study, which was conducted from May 2018 to January 2019, included community-dwelling adults in Taiwan aged ≥65 years. Stratified cluster sampling was performed, with seniors’ recreation centers selected randomly according to their location in urban, rural, and mountainous areas. Individuals were excluded if they had mental disorders or expressive language disorders, as indicated by the possession of integrated circuit card for severe illness; mild cognitive impairment or dementia, as determined using the Short Portable Mental Status Questionnaire (SPMSQ) [21]; and high-to-total dependence, according to the Activities of Daily Living (ADL) scale [22]. The final analysis included 1100 older adults (response rate: 93.2%). Post hoc power was computed using the effect size, calculated as the ratio of variance between depression status, the sample size and the α level was 0.05. The power was greater than 0.8, which indicated that the sample size was sufficiently large.

### 2.2. Instruments

Data were collected using a structured questionnaire developed by Lu et al. [23]. The questionnaire contains items on demographic information (age, sex, and education), depression status (as indicated by scores on the 15-item Geriatric Depression Scale (GDS-15) [24]), physical conditions (as indicated by chronic disease diagnosis, and scores on the Pittsburgh Sleep Quality Index (PSQI) [25], Study of Osteoporotic Fractures (SOF) index [26], SARC-F (Strength, assistance with walking, rising from a chair, climbing stairs, and falls) questionnaire [27], SPMSQ [21], and ADL scale [22]), and oral conditions (with reference to the Ohkuma Questionnaire [28] and the Xerostomia Inventory [29]). To assess content validity, a panel of experts reviewed all the items. The content validity index, which was between 0.89 and 1.00, was adequate. To ensure that the participants understood the content, the questionnaires were pilot tested on 30 older adults. The reliability of each scale was assessed according to internal consistency (Cronbach’s alpha).

### 2.3. Dental Examinations

Dental examinations were performed by seven dentists based on the World Health Organization criteria [30]. Regarding interrater agreement, the Kendall’s W for the Silness–Löe Plaque Index was 0.87, respectively. Concerning occlusal support, the participants’ Eichner Index (EI) was recorded. Furthermore, participants’ oral hygiene (including the Silness–Löe Plaque Index) was assessed.

### 2.4. Dependent Variable

#### Late-Life Depression

The GDS-15 is a self-reported measure of late-life depression in older adults. The 15 items were selected from the Long Form Geriatric Depression Scale because of their high correlation with depressive symptoms. Users respond in a “Yes/No” format. We administered the Chinese version of the GDS-15 [24]. Scores range 0 to 15, with scores greater than 5 suggesting depression. The overall Cronbach’s alpha was 0.80, the intra-class coefficient of the test–retest reliability over two weeks was 0.83, and the interrater reliability was 0.94 (intra-class) and 0.99 (Cohen’s kappa) [24].

### 2.5. Independent Variables

#### 2.5.1. Physical Conditions

Physical conditions included insomnia, physical frailty, sarcopenia, and comorbidities. All variable data were collected during the study and are listed as follows.

Insomnia

Insomnia was measured using the Chinese version of the PSQI, a self-rated questionnaire that assesses sleep quality and sleep disturbances. The Chinese version has been validated and determined to have adequate reliability [25]. The 19 items evaluate seven components of sleep quality, namely subjective sleep quality, sleep onset latency, total sleep duration, sleep efficiency, sleep disturbances, use of sleep medication, and daytime dysfunction. The subscore of each component ranges from 0 to 3, and the maximum total composite score is 21. The cutoff score for PSQI-defined cases of primary insomnia is ≥6.

Physical frailty

The SOF criteria are regarded as being as effective as the frailty criteria for predicting adverse health outcomes but are easier to apply [26]. The SOF index comprises three items (weight loss, chair stands, and energy level) with a score range of 0 to 3. One point is given for each of weight loss ≥5% in the past year, inability to complete five consecutive chair rises, and having no or little energy. Summed scores of 2 or 3, 1, and 0 correspond to frailty, pre-frailty, and robustness, respectively.

Sarcopenia

SARC-F has high specificity (94–99%) and is a suitable tool for community screening for sarcopenia [27]. The five SARC-F components are strength, assistance with walking, rising from a chair, climbing stairs, and falls. Total scores range from 0 to 10, with possible scores of 0 to 2 points for each component. A score of ≥4 is predictive of sarcopenia.

Comorbidities

This variable was identified by self-reports of more than two chronic diseases, such as hypertension, diabetes, heart disease, and chronic obstructive pulmonary disease.

#### 2.5.2. Oral Condition

Oral condition was examined across six components, namely dysphagia, xerostomia, masticatory performance, the EI (indicating occlusal support), oral diadochokinetic rate, and the Silness–Löe Plaque Index (representing oral hygiene). For each item, a score of 1 was defined as meeting the targeted measures as the presence of dysphagia, xerostomia, poor masticatory performance, EI categories B3–C3, oral diadochokinetic rate (indicated by failure to pronounce the “ta” monosyllable more than six times per second), and a Silness–Löe Plaque Index score >0.95. Total scores of 0, 1–2, and ≥3 points corresponded to non-oral frailty, pre-oral frailty, and oral frailty, respectively. The six components were investigated as follows.

Dysphagia

Dysphagia was assessed through rapid screening by using the Ohkuma questionnaire, which contains 15 items [28]. Example questions include “Do you ever have difficulty swallowing,” “Do you ever have difficulty with coughing up phlegm during or after a meal,” “Does it take you longer to eat a meal than before,” “Do you feel that it is becoming difficult to eat solid foods,” and “Do you ever have difficulty sleeping because of coughing during the night?” Possible responses are “obviously” (frequently), “slightly” (sometimes), and “no” (never). Respondents with at least one severe symptom were defined as having dysphagia. Regarding the internal consistency, the Cronbach’s alpha was 0.85.

Xerostomia

A condensed version of the Xerostomia Inventory was administered to identify and classify mouth dryness [29]. The following five items were used: “My mouth feels dry when I eat a meal,” “My mouth feels dry,” “I have difficulty eating dry foods,” “I have difficulty swallowing certain foods,” and “My lips feel dry.” Each item was assigned a score of 1 (never), 2 (occasionally), or 3 (often). Total scores ranged from 5 to 15 points, with a higher score reflecting a higher level of mouth dryness. Respondents with total scores of ≥10 points represented the top 50% of total xerostomia scores. Regarding the internal consistency, the Cronbach’s alpha was ≥0.70.

Masticatory performance

Masticatory performance was evaluated using color-changing chewing gum (Gum XYLITOL, Lotte, Tokyo, Japan). This chewing gum contains xylitol, citric acid, and red, yellow, and blue dyes that change color when subjected to masticatory forces from chewing. The red dye is pH sensitive and changes color under neutral or alkaline conditions. Citric acid maintains the internal pH of the initially yellowish-green gum at a low level before chewing commences. As chewing progresses, the yellow and blue dyes seep into the saliva, and the release of citric acid causes the gum to turn red [31]. Participants were asked to chew the gum for 2 min, after which an observer checked the color of the gum by using a color chart of five color gradations ranging from 1 (very poor) to 5 (very good). For statistical purposes, we classified masticatory performance into three categories: 1–3 = poor, 4 = moderate, and 5 = good. Kamiyama et al. [32] confirmed the validity and reliability of using color-changing chewing gum to evaluate masticatory performance.

Occlusal support

Occlusal support was evaluated using the EI [33], which is based on an individual’s number of POSAs. The EI categories A1–A3, B1–B4, and C1–C3 indicate contact in four POSAs; three, two, one, or zero POSAs; and zero POSAs, respectively. According to Lin et al. [16], the preservation of at least two POSAs is vital for adequate masticatory function; thus, we simplified EI into two categories: A1–B2 and B3–C3.

Oral diadochokinesis rate

Oral diadochokinesis was assessed by examining articulatory oral motor skills at sites such as the lips, tip of the tongue, and the dorsum of the tongue [34]. The participants were instructed to repeatedly pronounce the monosyllables “pa” “ta” and “ka” for 5 s as quickly as possible, and the number of syllables pronounced per second was noted. Oral diadochokinetic was calculated separately for each syllable as the articulation count per second and categorized as <6 or ≥6 times per second.

Oral hygiene

Measurement of the state of oral hygiene by using the Silness–Löe Plaque Index is based on the examination of both soft debris and mineralized deposits on the number 12, 16, 24, 32, and 44 teeth. Each of the four surfaces of the teeth (buccal, lingual, mesial, and distal) is given a score from 0 to 3.

### 2.6. Data Collection

Data were collected through face-to-face interviews, which were administered by well-trained interviewers in compliance with a standard protocol. The collection process comprised three steps lasting an hour total. First, a dentist performed the dental examination and recorded the dental status. Second, an interviewer administered the structured questionnaire. The entire interview process took approximately 30 to 45 min. Finally, the research personnel collected masticatory performance data, recorded the monosyllable pronunciation data, and physical function as complete five consecutive chair rises, walking 5 m, climbing a flight of 10 stairs, lifting and carrying 4.5 kg.

### 2.7. Statistical Analysis

The participants’ depression status was categorized as late-life depression or non-late-life depression. The data are expressed as means and standard deviations or as frequencies and percentages, and the two-sample t test and χ^2^ test were conducted to assess the relationship between the factors (continuous and categorical) and outcomes. Variables exhibiting statistically significant associations in univariate analysis were included in the multivariate analysis. The adjusted odds ratio (aOR) and 95% confidence interval (CI) obtained through the exponentiation of the corresponding regression coefficient were employed in evaluating the association between the study variables and depression status after the effects of potential confounders (sex, age, and education) were adjusted for. Multivariable logistical regression models were used to evaluate the combined effects of physical frailty and oral frailty on depression status.

## 3. Results

### 3.1. Participant Characteristics

The characteristics of the participants, categorized according to age, are presented in Table 1. Women and men constituted 71.4% and 28.6% of the 1100 participants, respectively. In total, 9.3% of the participants had late-life depression, and no difference between age group was noted in this regard. Significantly higher percentages of older adults aged ≥75 years had physical frailty, sarcopenia, and comorbidities (3%, 10%, and 34.1%, respectively) compared with participants in the group of older adults aged ≤74 years (1.3%, 1.2%, and 28.3%, respectively). Furthermore, the percentages of older adults aged ≥75 years with pre-oral frailty and oral frailty (60.4% and 30.2%, respectively) were significantly greater than those in the ≤74 years (55.2% and 15%, respectively). Rates of poor masticatory performance, dysphagia, and xerostomia were significantly higher among the older adults aged ≥75 years than among those aged ≤74 years (*p* < 0.001). Of the older adults aged ≥75 years, 62.3% had an occlusal condition in the B3–C3 category, and the mean of the Silness–Löe Plaque Index was 1.04. Significantly higher proportions of older adults aged ≥75 years (33%, 34.5%, and 33%, respectively) failed to pronounce the “pa” “ta” and “ka” monosyllables six or more times per second compared with those in the ≤74 years (13.3%, 16%, and 16.4%, respectively).

### 3.2. Physical and Oral Conditions Associated with Late-Life Depression

Table 2 displays the results of the multiple regression analysis for the associations of physical frailty and oral frailty with late-life depression. Even after adjustment for variables, older adults with insomnia and sarcopenia were 2.56 and 4.25 times more likely to develop late-life depression than were their unaffected counterparts, respectively. Factors associated with late-life depression were pre-physical frailty (aOR = 3.61, 95% CI: 1.94–6.71), physical frailty (aOR = 53.74, 95% CI: 12.87–224.42), pre-oral frailty (aOR = 2.56, 95% CI: 1.12–5.84), and oral frailty (aOR = 4.89, 95% CI: 2.02–11.80). Moreover, physical frailty and oral frailty were associated with late-life depression in a dose–response manner (*p* for trend <0.001).

Table 3 presents the associations between six components of oral frailty and late-life depression. The participants with dysphagia were more likely to have than to not have late-life depression (aOR = 2.85, 95% CI: 1.55–5.23) and xerostomia (aOR = 1.10, 95% CI: 1.01–1.22).

### 3.3. Combined Effects of Physical and Oral Conditions on Late-Life Depression

Table 4 displays the combined effects of pre-physical frailty and oral frailty (aOR = 8.23, 95% CI: 3.80–17.79), physical frailty and oral frailty (aOR = 36.81, 95% CI: 5.71–237.27), sarcopenia and oral frailty (aOR = 8.60, 95% CI: 2.84–25.97), and insomnia and oral frailty (aOR = 6.21, 95% CI: 2.69–14.34).

Table 5 presents the combined effects of physical frailty and dysphagia (aOR = 152.31, 95% CI: 17.60–1317.96), physical frailty and xerostomia (aOR = 67.85, 95% CI: 12.75–360.90), sarcopenia and dysphagia (aOR = 21.29, 95% CI: 8.31–54.52), sarcopenia and xerostomia (aOR = 18.23, 95% CI: 7.03–47.29), insomnia and dysphagia (aOR = 12.41, 95% CI: 6.22–24.76), and insomnia and xerostomia (aOR = 10.98, 95% CI: 5.12–23.52).

## 4. Discussion

The results confirm the premise that physical frailty and oral frailty are associated with late-life depression. Notably, physical frailty, sarcopenia, insomnia, oral frailty, dysphagia, and xerostomia exerted significant combined effects in the participants with all these conditions.

Older adults with physical frailty, sarcopenia, and insomnia were more likely to have late-life depression than were older adults without all these conditions. The rates of physical frailty and sarcopenia in the participants aged ≥75 years were three and nine times the corresponding rates in the participants aged ≤74 years. Sarcopenia and depressive symptoms are associated with frailty [7]. Azzolino et al. [8] reported that aging is accompanied by several changes that may affect swallowing function, termed presbyphagia. Moreover, the age-related decline in skeletal muscle is systemic; sarcopenia exerts a negative effect on the swallowing muscles, such as the geniohyoid and tongue muscles, which contributes to the onset of sarcopenic dysphagia [9]. In sum, the physical and oral conditions of older adults are closely correlated.

Pre-oral frailty and oral frailty were associated with late-life depression in a dose–response manner. Oral frailty was twice as prevalent among the older adults aged 75 and older compared with among those aged 65 to 74 years, indicating that oral function declines with age. This was also found in previous research. The researchers found a turning point in oral health from the age of 75 [35]. Eating and swallowing are complex oral food processing behaviors divided into preparation, oral, pharyngeal, and esophageal stages depending on the location of the bolus [36]. In the oral stage, food is formed into a bolus and requires mastication, the application of tongue pressure, and salivation before it can be swallowed. Poor masticatory performance and xerostomia cause difficulty in bolus formation [37], and poor tongue strength makes it challenging to wrap the tongue around the bolus and move it to the pharynx [36]. These impaired oral functions also interfere with other swallowing efforts. Ketel et al. [17] noted that aging-related changes in oral physiology can affect oral food processing.

We observed the participants with both dysphagia and xerostomia were more likely to have late-life depression. Polypharmacy is common among older adults. Medication-induced xerostomia was higher with polypharmacy, most notably in those taking antidepressants [14]. In the cohort study, patients with swallowing problems had documentation of a concomitant xerostomia diagnosis. [38]. Studies have noted strong associations of depression with dysphagia, perceived oral health, a higher number of missing teeth, and the prevalence of xerostomia among older adults [11,12]. Poor oral hygiene is a risk factor for periodontitis. Severe periodontitis cause tooth loss is a common problem among older adults. One study reported that improve periodontal care practices and receive proper periodontal treatment, if necessary, to improve their oral health-related quality of life [39,40].

Examination of combined effects revealed that the participants with both physical and oral conditions such as physical frailty and dysphagia, or physical frailty and xerostomia, are at increased risk of developing late-life depression. Azzolino et al. [8] identified the that sarcopenia and dysphagia are common among older adults and are closely correlated. Herein, a combined effects analysis indicated that the participants with both sarcopenia and dysphagia were associated with late-life depression.

This study has some limitations. First, because the sample comprised community-dwelling older adults from southern Taiwan, the findings may not be generalizable to all older adults. However, the study methodology can be extended to investigations of community-dwelling older adults’ physical and oral function. Second, we did not evaluate lips and tongue muscle strength, which is related to swallowing motion. However, oral diadochokinetic can be applied to evaluate oral motor skills at sites such as the lips and the tip and dorsum of the tongue in community-dwelling older adults. Third, we used subjective Xerostomia Inventory to identify and classify mouth dryness rather than objective instruments to measure xerostomia. However, the scale is a valid measure for discriminative use in clinical and epidemiologic research [29]. Fourth, participants self-reported the number of chronic diseases rather than medications. This may not take into account the medication-induced xerostomia. Finally, the cross-sectional nature of this study precludes the inference of causal relationships among the variables.

## 5. Conclusions

Physical frailty and oral frailty were significantly associated with late-life depression in community-dwelling older adults in a dose-response manner. Physical frailty, sarcopenia, insomnia, oral frailty, dysphagia, and xerostomia exerted significant combined effects on late-life depression. The findings suggest that developing early intervention strategies is integral to prevent frailty among older adults, which can in turn reduce the likelihood of late-life depression onset among them.

## Figures and Tables

**Table 1 jpm-12-00459-t001:** The characteristics of the participants.

Variables	Total (*n* = 1100)		≤74 y Old (*n* = 639)		≥75 y Old (*n* = 461)		*p* Value
	*n*	(%)	*n*	(%)	*n*	(%)	
Demographic							
Sex							0.088
Men	314	(28.6)	195	(30.5)	119	(25.8)	
Women	786	(71.4)	444	(69.5)	342	(74.2)	
Education							<0.001
Above college	225	(20.4)	180	(28.2)	45	(9.8)	
Junior/High school	369	(33.6)	247	(38.6)	122	(26.5)	
Illiterate/Elementary	506	(46.0)	212	(33.2)	294	(63.8)	
Late-life depression							0.123
No	995	(90.7)	586	(91.8)	409	(89.1)	
Yes	102	(9.30)	52	(8.20)	50	(10.9)	
Physical conditions							
Insomnia							0.402
No	635	(65.5)	374	(64.5)	261	(67.1)	
Yes	334	(34.5)	206	(35.5)	128	(32.9)	
Physical frailty							0.002
Non	863	(78.5)	524	(82.0)	339	(73.5)	
Pre-frailty	215	(19.6)	107	(16.7)	108	(23.4)	
Frailty	22	(2.0)	8	(1.3)	14	(3.0)	
Sarcopenia							<0.001
No	1046	(95.1)	631	(98.8)	415	(90.0)	
Yes	54	(4.9)	8	(1.2)	46	(10.0)	
Comorbidities							0.042
No	762	(69.3)	458	(71.7)	304	(65.9)	
Yes	338	(30.7)	181	(28.3)	157	(34.1)	
Oral conditions							
Oral frailty status							<0.001
Non	201	(22.1)	169	(29.8)	32	(9.4)	
Pre-frailty	519	(57.2)	313	(55.2)	206	(60.4)	
Frailty	188	(20.7)	85	(15.0)	103	(30.2)	
Occlusal support							
Eichner index							<0.001
A1 to B2	569	(51.7)	395	(61.8)	174	(37.7)	
B3 to C3	531	(48.3)	244	(38.2)	387	(62.3)	
Masticatory performance							<0.001
Good/moderate	792	(72.9)	515	(81.6)	277	(60.7)	
Poor	295	(27.1)	116	(18.4)	179	(39.3)	
Dysphagia							<0.001
No	952	(86.6)	574	(89.8)	378	(82.0)	
Yes	148	(13.5)	65	(10.2)	83	(18.0)	
Xerostomia							<0.001
No	972	(88.4)	589	(92.2)	383	(83.1)	
Yes	128	(11.6)	50	(7.8)	78	(16.9)	
Oral hygiene							
Plaque Index (mean ± SD)	0.94 ± 0.02		0.88 ± 0.52		1.04 ± 0.55		<0.001
Oral diadochokinesis rate							
“pa”							<0.001
≥6 (times/sec)	863	(78.5)	554	(86.7)	309	(67.0)	
<6 (times/sec)	237	(21.5)	85	(13.3)	152	(33.0)	
“ta”							<0.001
≥6 (times/sec)	839	(76.3)	537	(84.0)	302	(65.5)	
<6 (times/sec)	261	(23.7)	102	(16.0)	159	(34.5)	
“ka”							<0.001
≥6 (times/sec)	843	(76.6)	534	(83.6)	309	(67.0)	
<6 (times/sec)	257	(23.4)	105	(16.4)	152	(33.0)	

Notes: y, years; SD, standard deviation.

**Table 2 jpm-12-00459-t002:** Odds ratio for late-life depression associated with physical conditions and oral frailty.

Variables	Late-Life Depression							
	No (*n* = 995)		Yes (*n* = 102)		cOR	(95% CI)	aOR	(95% CI)
	*n*	(%)	*n*	(%)				
Demographic								
Age								
≤74 y old (ref.)	586	(58.9)	52	(51.0)	1.00		1.00	
≥75 y old	409	(41.1)	50	(49.0)	1.37	(0.91–2.07)	1.02	(0.53–1.95)
Sex								
Male (ref.)	286	(28.7)	26	(25.5)	1.00		1.00	
Female	709	(71.3)	76	(74.5)	1.18	(0.73–1.87)	0.93	(0.44–1.93)
Education								
Above college (ref.)	207	(20.8)	17	(16.7)	1.00		1.00	
Junior/High school	341	(34.3)	26	(25.5)	0.93	(0.49–1.75)	0.88	(0.35–2.19)
Illiterate/Elementary	447	(44.9)	59	(57.8)	1.60	(0.91–2.82)	1.21	(0.47–3.08)
Physical conditions								
Insomnia								
No (ref.)	604	(67.9)	29	(37.7)	1.00		1.00	
Yes	286	(32.1)	48	(62.3)	2.78	(2.02–3.79)	2.56	(1.41–4.67)
Physical frailty								
Non (ref.)	819	(82.3)	41	(40.2)	1.00		1.00	
Pre-frailty	170	(17.1)	45	(44.1)	5.28	(3.35–8.32)	3.61	(1.94–6.71)
Frailty	6	(0.6)	16	(16.7)	53.26	(19.80–143.25)	53.74	(12.87–224.42)
*p for trend*					<0.001		<0.001	
Sarcopenia								
No (ref.)	961	(96.6)	82	(80.4)	1.00		1.00	
Yes	34	(3.4)	20	(19.6)	6.89	(3.79–12.51)	4.25	(1.64–11.01)
Comorbidities								
No (ref.)	704	(70.8)	57	(55.9)	1.00		1.00	
Yes	291	(29.3)	45	(44.1)	1.90	(1.26–2.89)	1.33	(0.73–2.44)
Oral frailty status								
Non (ref.)	193	(23.4)	7	(8.5)	1.00		1.00	
Pre-frailty	474	(57.5)	45	(54.9)	2.61	(1.16–5.90)	2.56	(1.12–5.84)
Frailty	157	(19.1)	30	(36.6)	5.26	(2.25–12.31)	4.89	(2.02–11.80)
*p for trend*					<0.001		<0.001	

Notes: cOR = crude odds ratio; aOR = adjusted odds ratio; 95% CI = 95% confidence interval; ref. = reference group; aORs were adjusted for variables in the table.

**Table 3 jpm-12-00459-t003:** Odds ratio for late-life depression associated with six components of oral frailty.

Variables	Late-Life Depression							
	No (*n* = 955)		Yes (*n* = 102)		cOR	(95% CI)	aOR	(95% CI)
	*n* (%)		*n* (%)					
Occlusal support								
Eichner index								
A1 to B2 (ref.)	519	(52.2)	49	(48.0)	1.00		-	
B3 to C3	476	(47.8)	53	(52.0)	1.17	(0.78–1.77)	-	
Masticatory performance								
Good/Moderate (ref.)	728	(74.1)	63	(61.8)	1.00		1.00	
Poor	254	(25.9)	39	(38.2)	1.77	(1.16–2.71)	1.16	(0.63–2.14)
Dysphagia								
No (ref.)	888	(89.3)	62	(60.8)	1.00		1.00	
Yes	107	(10.7)	40	(39.2)	5.35	(3.43–8.35)	2.85	(1.55–5.23)
Xerostomia								
No (ref.)	900	(90.5)	69	(67.6)	1.00		1.00	
Yes	95	(9.6)	33	(32.4)	4.53	(2.84–7.21)	1.10	(1.01–1.22)
Oral hygiene								
Plaque Index								
≤0.94 (ref.)	420	(50.4)	36	(43.9)	1.00		-	
>0.95	413	(49.6)	46	(56.1)	1.30	(0.82–2.05)	-	
Oral disdochokinesis rate								
“ta”								
≥6 (times/sec) (ref.)	767	(77.1)	69	(67.7)	1.00		1.00	
<6 (times/sec)	228	(22.9)	33	(32.3)	1.61	(1.04–2.49)	1.28	(0.67–2.42)

Notes: cOR = crude odds ratio; aOR = adjusted odds ratio; 95% CI = 95% confidence interval; ref. = reference group; aORs were adjusted for age, sex, education, insomnia, physical frailty, sarcopenia, comorbidities as well as variables in the table.

**Table 4 jpm-12-00459-t004:** Combined effects of physical conditions and oral frailty status for late-life depression.

Variables	Variables		Total		Late-Life Depression		
			*n*	(%)	*n*	aOR	(95% CI)
Physical frailty	Oral frailty						
Non	Non + Pre-frailty	(ref.)	587	(64.6)	22	1.00	
Non	Frailty		127	(14.0)	13	2.86	(1.38–5.95)
Pre-frailty	Non + Pre-frailty		122	(13.4)	22	5.64	(3.00–10.60)
Pre-frailty	Frailty		56	(6.2)	14	8.23	(3.80–17.79)
Frailty	Non + Pre-frailty		11	(1.2)	8	69.21	(16.99–281.93)
Frailty	Frailty		5	(0.6)	3	36.81	(5.71–237.27)
Sarcopenia	Oral frailty						
No	Non + Pre-frailty	(ref.)	697	(76.8)	45	1.00	
No	Frailty		172	(18.9)	24	2.29	(1.32–3.97)
Yes	Non + Pre-frailty		23	(2.5)	7	6.39	(2.41–16.94)
Yes	Frailty		16	(1.8)	6	8.60	(2.84–25.97)
Insomnia	Oral frailty						
No	Non + Pre-frailty	(ref.)	432	(53.6)	16	1.00	
No	Frailty		100	(12.4)	11	2.64	(1.13–6.18)
Yes	Non + Pre-frailty		216	(26.9)	26	3.95	(2.02–7.71)
Yes	Frailty		57	(7.1)	12	6.21	(2.69–14.34)

Notes: aOR = adjusted odds ratio; 95% CI = 95% confidence interval; ref. = reference group; aORs were adjusted for age, sex, and education.

**Table 5 jpm-12-00459-t005:** Combined effects of physical conditions, dysphagia, and xerostomia for late-life depression.

Variables	Variables		Total		Late-Life Depression		
			*n*	(%)	*n*	aOR	(95% CI)
Physical frailty	Dysphagia						
Non	No	(ref.)	778	(70.7)	29	1.00	
Non	Yes		85	(7.7)	12	4.36	(2.12–8.99)
Pre-frailty	No		159	(14.5)	23	4.43	(2.47–7.91)
Pre-frailty	Yes		56	(5.1)	22	16.59	(8.53–32.23)
Frailty	No		15	(1.4)	10	51.88	(16.45–163.60)
Frailty	Yes		7	(0.6)	6	152.31	(17.60–1317.96)
Physical frailty	Xerostomia						
Non	No	(ref.)	793	(72.1)	32	1.00	
Non	Yes		70	(6.4)	9	3.44	(1.55–7.62)
Pre-frailty	No		165	(15.0)	27	4.67	(2.69–8.09)
Pre-frailty	Yes		50	(4.5)	18	13.36	(6.71–26.63)
Frailty	No		14	(1.3)	10	59.28	(17.46–201.30)
Frailty	Yes		8	(0.7)	6	67.85	(12.75–360.90)
Sarcopenia	Dysphagia						
No	No	(ref.)	919	(83.5)	56	1.00	
No	Yes		127	(11.5)	29	4.55	(2.73–7.57)
Yes	No		33	(3.0)	8	5.16	(2.15–12.37)
Yes	Yes		21	(1.9)	12	21.29	(8.31–54.52)
Sarcopenia	Xerostomia						
No	No	(ref.)	938	(85.3)	60	1.00	
No	Yes		108	(9.8)	22	3.73	(2.16–6.45)
Yes	No		34	(3.1)	9	5.40	(2.32–12.57)
Yes	Yes		20	(1.8)	11	18.23	(7.03–47.29)
Insomnia	Dysphagia						
No	No	(ref.)	576	(59.4)	19	1.00	
No	Yes		59	(6.1)	10	5.50	(2.39–12.63)
Yes	No		265	(27.3)	26	3.32	(1.79–6.15)
Yes	Yes		69	(7.2)	22	12.41	(6.22–24.76)
Insomnia	Xerostomia						
No	No	(ref.)	590	(60.9)	21	1.00	
No	Yes		45	(4.6)	10	7.49	(3.19–17.56)
Yes	No		281	(29.0)	33	4.16	(2.30–7.52)
Yes	Yes		53	(5.5)	15	10.98	(5.12–23.52)

Notes: aOR = adjusted odds ratio; 95% CI = 95% confidence interval; ref. = reference group; aORs were adjusted for age, sex, and education.

## Data Availability

The data that support the findings of this study are available from the corresponding author upon reasonable request.

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
