# Peer review of "Physical Frailty and Oral Frailty Associated with Late-Life Depression in Community-Dwelling Older Adults"

_jpm, 2022, doi:10.3390/jpm12030459_

Round 1

Reviewer 1 Report

Dear authors,

My compliments to the number of community-dwelling older people included in this study and the large amount of data that has been collected.

However, I also have comments to improve you article.

General comments:

- In my opinion, insufficient attention is paid to the concept of frailty. A commonly used definition is that of Gobbens: Frailty, defined as a process of accumulation of physical, psychological and/or social deficits in functioning, increasing the risk of negative health outcomes such as disabilities, admission to health care facilities, and death, is increasingly common in older people (Gobbens et al, 2012). The different domains of frailty are closely interrelated. And when a deficiency arises in one domain, it often has a direct effect on another domain.

- I believe that oral frailty in the elderly cannot be 'a condition that manifests itself only in the oral cavity'. Because deterioration in oral health in the elderly is (almost) always related to somatic disorders and polypharmacy.

And the conclusion that developing interventions to improve oral function would have an effect on the prevalence of depression is too short-sighted. Oral function may have deteriorated because of a poorer lifestyle. That lifestyle has also led to cardiovascular disease, which results in polypharmacy. So a relationship between physical frailty and oral frailty is obvious. The association between oral frailty and depression can be explained in many ways.

1. Introduction:

- I miss possible causes of late-life depression from the literature

- l.48 The word 'physiconutritional' deserves explanation.

- l.51 Mention the risk of aspiration and aspiration pneumonia

- The last paragraph should be formulated with a clear research question. The last sentence does not belong in the introduction. Or should be phrased as 'If there is an association between... early interventions might prevent....'.

2.M&M:

What are community care centres? Do people from the neighbourhood spend their day here or do they come here for care?

L83-84. Individuals were excluded if they had mental disorders', other than dementia?

2.5.      Independent variables

I miss the number of medications and whether hyposalivation inducing medications such as antidepressants are used. Many older people do not experience xerostomia, but hyposalivation is often present. For assessment of oral frailty, polypharmacy and hyposalivation seem essential. Moreover, hyposalivation is related to dysphagia.

4. Discussion:

- It is known that oral frailty is related to age, and also that there is a turning point. See also the publication: Oral Health of Older Patients in Dental Practice: An Exploratory Study - PubMed (nih.gov)

- ‘We observed that oral frailty was associated with late-life depression'. What role do antidepressants play in this relationship?

- ‘Participants with both dysphagia and xerostomia were more likely to have late-life depression'. Again, medication may play a role.

- Do not repeat results in the discussion.

- Conclusion: late-life depression is part of frailty, see definition by Gobbens.

- I can confirm that early physical interventions can contribute to improvements in lifestyle and that this can also have a positive effect on oral health and late-life depression (stopping smoking reduces the prevalence of cardiovascular diseases and periodontal disease).

Author Response

Dear authors,

My compliments to the number of community-dwelling older people included in this study and the large amount of data that has been collected.

However, I also have comments to improve your article.

General comments:

Point 1: In my opinion, insufficient attention is paid to the concept of frailty. A commonly used definition is that of Gobbens: Frailty, defined as a process of accumulation of physical, psychological and/or social deficits in functioning, increasing the risk of negative health outcomes such as disabilities, admission to health care facilities, and death, is increasingly common in older people (Gobbens et al, 2012). The different domains of frailty are closely interrelated. And when a deficiency arises in one domain, it often has a direct effect on another domain.

Response 1: We have made correction according to the Reviewer’s comments, and cited the Gobbens et al (2012) in the 2nd paragraph of introduction. (P.2 Line 46-49)

- Gobbens, R.J.; van Assen, M.A.; Luijkx, K. G.; Schols, J.M. Testing an integral conceptual model of frailty. J. Adv. Nurs. 2012, 68, 2047-2060.

Point 2: I believe that oral frailty in the elderly cannot be 'a condition that manifests itself only in the oral cavity'. Because deterioration in oral health in the elderly is (almost) always related to somatic disorders and polypharmacy.

 Response 2: It is true as Reviewer’s comment, and we modified the 3rd paragraph of introduction. (P.2 Line 61-63)

-Thomson, W.M.; Ferguson, C.A.; Janssens, B.E.; Kerse, N.M.; Ting, G.S.; Smith, M.B. Xerostomia and polypharmacy among dependent older New Zealanders: a national survey. Age Ageing 2021, 50, 248-251.

Point 3: And the conclusion that developing interventions to improve oral function would have an effect on the prevalence of depression is too short-sighted. Oral function may have deteriorated because of a poorer lifestyle. That lifestyle has also led to cardiovascular disease, which results in polypharmacy. So a relationship between physical frailty and oral frailty is obvious. The association between oral frailty and depression can be explained in many ways.

 Response 3: As Reviewer’s comments, we have modified the conclusion “…developing early intervention strategies is integral to prevent frailty among older adults….” (P.10 Line 344-346)

  1. Introduction:

Point 4: I miss possible causes of late-life depression from the literature

 Response 4: As Reviewer suggested that added the literature of the causes of late-life depression. (P.1 Line 44)

- Blazer, D.G. Depression in late life: review and commentary. J. Gerontol. A Biol. Sci. Med. Sci. 2003, 58, 249-265.

Point 5: l.48 The word 'physiconutritional' deserves explanation.

Response 5: Jung et al. (2020) defined the word “physiconutritional” as following: having ≥1 risk of malnutrition, sarcopenia, severe mobility limitation, longer Timed Up and Go (>12 s), and low Short Physical Performance Battery (≤9 scores). We have added the definitions of “physiconutritional”in the paragraph. (P.2 Line 50)

- Jung, H.; Kim, M.; Lee, Y.; Won, C.W. Prevalence of physical frailty and its multidimensional risk factors in Korean commu-nity-dwelling older adults: findings from Korean frailty and aging cohort study. Int. J. Environ. Res. Public Health 2020, 17, 7883.

Point 6: l.51 Mention the risk of aspiration and aspiration pneumonia

Response 6: Thanks for the suggestion, the literature of the risk of aspiration and aspiration pneumonia was added, and highlight with red. (P.2 Line 54-56)

-Wakabayashi, H. Presbyphagia and sarcopenic dysphagia: association between aging, sarcopenia, and deglutition disorders. J. Frailty Aging 2014, 3, 97-103.

Point 7: The last paragraph should be formulated with a clear research question. The last sentence does not belong in the introduction. Or should be phrased as 'If there is an association between... early interventions might prevent....'.

 Response 7: We have re-written this part according to the Reviewer’s suggestion, and deleted the last sentence in the introduction.

  1. M&M:

Point 8: What are community care centres? Do people from the neighbourhood spend their day here or do they come here for care?

Response 8: Community care center is like the seniors’ recreation center. Older adults would get variety of programs and services on day time at the center. The word “seniors’ recreation center “instead of “Community care center “(P.2 Line 81)

Point 9: L83-84. Individuals were excluded if they had mental disorders', other than dementia?

 Response 9: Individuals were excluded if they had …“dementia, as determined using the Short Portable Mental Status Questionnaire (SPMSQ)”…was added.(P.2 Line 84-85)

2.5. Independent variables

Point 10: I miss the number of medications and whether hyposalivation inducing medications such as antidepressants are used. Many older people do not experience xerostomia, but hyposalivation is often present. For assessment of oral frailty, polypharmacy and hyposalivation seem essential. Moreover, hyposalivation is related to dysphagia.

 Response 10: Participants self-reported the number of chronic diseases rather than medications. Therefore, the medication-induced xerostomia was a bias in this study. We have added this issue into study limitation. (P.10 Line 336-338)

  1. Discussion:

Point 11: It is known that oral frailty is related to age, and also that there is a turning point. See also the publication: Oral Health of Older Patients in Dental Practice: An Exploratory Study - PubMed (nih.gov)

 Response 11: Thanks for providing the valuable reference. We have cited this article in the main text. (P.10 Line 300)

-Bots-VantSpijker, P.C.; van der Maarel-Wierink, D.C.; Schols, J.M.; Bruers, J.J. Oral Health of Older Patients in Dental Prac-tice: An Exploratory Study. Int. Dent. J. 2021, 5, S0020-6539(21)00099-X.

Point 12: ‘We observed that oral frailty was associated with late-life depression'. What role do antidepressants play in this relationship? - ‘Participants with both dysphagia and xerostomia were more likely to have late-life depression'. Again, medication may play a role.

 Response 12: We have added the literatures in the discussion. “Polypharmacy is common among older adults. Medication-induced xerostomia was higher with polypharmacy, most notably in those taking antidepressants. Moreover, dysphagia in the older adults is related to somatic disorders.”(P.10 Line 310-312)

-Thomson, W.M.; Ferguson, C.A.; Janssens, B.E.; Kerse, N.M.; Ting, G.S.; Smith, M.B. Xerostomia and polypharmacy among dependent older New Zealanders: a national survey. Age Ageing 2021, 50, 248-251.

-Azzolino, D.; Damanti, S.; Bertagnoli, L.; Lucchi, T.; Cesari, M. Sarcopenia and swallowing disorders in older people. Aging Clin. Exp. Res. 2019, 31, 799-805.

Point 13: Do not repeat results in the discussion.

 Response 13: Considering the Reviewer’s suggestion, we have removed repeat results in the discussion and modified the description.  

Point 14: Conclusion: late-life depression is part of frailty, see definition by Gobbens.

Response14: It is really true as Reviewer’s comments. Gobbens et al. (2012) reported frailty as a process of accumulation of physical, psychological and/or social deficits in functioning.

- Gobbens, R.J.; van Assen, M.A.; Luijkx, K. G.; Schols, J.M. Testing an integral conceptual model of frailty. J. Adv. Nurs. 2012, 68, 2047-2060.

Point 15: I can confirm that early physical interventions can contribute to improvements in lifestyle and that this can also have a positive effect on oral health and late-life depression (stopping smoking reduces the prevalence of cardiovascular diseases and periodontal disease).

 Response 15: We agreed Reviewer’s comments. Those comments are all valuable and very helpful for revising and improving our paper, as well as the important guiding significance to our researches. We have studied comments carefully and have made correction which we hope meet with approval.

Reviewer 2 Report

The research question is relevant to the concept of Oral and Physical Frailty and the results and findings are consistent with the objectives. Association of physical and oral frailty with depression likely to distort, wherein depression would be a confounding factor in older adults. Establishing the temporal sequence of oral, physical frailty to depression cannot be fully elucidated.   

Author Response

Point 1: The research question is relevant to the concept of Oral and Physical Frailty and the results and findings are consistent with the objectives. Association of physical and oral frailty with depression likely to distort, wherein depression would be a confounding factor in older adults. Establishing the temporal sequence of oral, physical frailty to depression cannot be fully elucidated.

 Response 1: It is really true as Reviewer comment. This study investigated the associations of physical frailty and oral frailty with depression in older adults. Because of the cross-sectional nature of this study, causal inferences were not drawn. We have included this issue to study limitation. (P.10 Line 338-339)

Reviewer 3 Report

in Data collection section, the author mention about collecting data about physical function. Please add some information about wihich test did  you use to measure physical function

in Method section the author using  subjective method to measure  xerostomia. pleae add information  about why  you  didnt  use objective method  to measure xerostomia

Author Response

Point 1: in Data collection section, the author mention about collecting data about physical function. Please add some information about which test did you use to measure physical function

Response 1: Considering the Reviewer’s suggestion, we have added the test of physical function in the “2.6. Data collection” section and highlight in red. (P.5 Line 213-214)

Point 2: in Method section the author using subjective method to measure xerostomia. please add information about why you didn’t use objective method to measure xerostomia.

Response 2: A large-scale community survey uses convenient and economical summated rating scale to identify and participants’ situation. Thomson et al. (2011) reported the Summated Xerostomia Inventory is valid for measuring xerostomia symptoms in clinical and epidemiologic research. This study included 1100 community-dwelling older adults. Therefore, the Xerostomia Inventory was administered to identify and classify mouth dryness. We have included this issue to study limitation. (P.10 Line 333-336) 

-Thomson, W.M.; van der Putten, J.G.; de Baat, C.; Ikebe, K.; Matsuda, K.I.; Enoki, K.; Hopcraft, M.S.; Ling, G.Y. Shortening the xerostomia inventory. Oral Surg. Oral Med. Oral Pathol. Oral Radiol. Endod. 2011, 112, 322-327.

Reviewer 4 Report

The study aimed to assess the associations of physical and oral frailty with depression in 1100 institutionalized Taiwanese older adults. The study is relevant and the message to the reader is clear. I did not find any particular issue with this manuscript and I, therefore, suggest proceeding with the publication process.

Author Response

Point 1: The study aimed to assess the associations of physical and oral frailty with depression in 1100 institutionalized Taiwanese older adults. The study is relevant and the message to the reader is clear. I did not find any particular issue with this manuscript and I, therefore, suggest proceeding with the publication process.

Response 1: Thanks to you for your good comments.

Round 2

Reviewer 1 Report

Dear authors,

The manuscript has clearly improved. A few more comments:

  • The abstract still needs to be edited, especially the conclusion.
  • line 61-63: 'Deterioration in oral health in the elderly is most often related to somatic disorders and polypharmacy [14]. Medication-induced xerostomia is common among older adults, especially when they take antidepressants [14]. 
  • line 302: 'This was also found in previous research. The researchers found a turning point in oral health from the age of 75.'
  • line 314 'Moreover, dysphagia in the older adults is related to somatic disorders [8].'  This is not entirely correct. Xerostomia or hyposalivation can also cause dysphagia. 
  • I agree with the conclusion, please incorporate it into abstract as well.

Author Response

Response to Reviewer 1 Comments

Dear authors,

The manuscript has clearly improved. A few more comments:

Point 1: The abstract still needs to be edited, especially the conclusion.

Response 1: Thank you for the suggestion and the revised portion were marked in red in the paper.(Line 28-29)

Point 2: line 61-63: 'Deterioration in oral health in the elderly is most often related to somatic disorders and polypharmacy [14]. Medication-induced xerostomia is common among older adults, especially when they take antidepressants [14].

Response 2: We have made correction according to the Reviewer’s comment. (Line 61-63)

Point 3: line 302: 'This was also found in previous research. The researchers found a turning point in oral health from the age of 75.'

Response 3: We have modified the sentence to “This was also found in previous research….”. (Line 300-301)

Point 4: line 314 'Moreover, dysphagia in the older adults is related to somatic disorders [8].'  This is not entirely correct. Xerostomia or hyposalivation can also cause dysphagia.

Response 4: We have revised this portion to “In the cohort study, patients with swallowing problems had documentation of a concomitant xerostomia diagnosis [38]”. (Line 313-314)

- Marcott, S.; Dewan, K.; Kwan, M.; Baik, F.; Lee, Y. J.; Sirjani, D. Where dysphagia begins: polypharmacy and xerostomia. Federal Practitioner 2020, 37, 234.

Point 5: I agree with the conclusion, please incorporate it into abstract as well.

Response 5: We are grateful to you for the positive comments. The conclusion in abstract was revised.